# UNDERSTANDING GRAPH LEARNING WITH LOCAL INTRINSIC DIMENSIONALITY

## ABSTRACT

Many real-world problems can be formulated as graphs and solved by graph learning techniques. Whilst the rise of Graph Neural Networks (GNNs) has greatly advanced graph learning, there is still a lack of understanding of the intrinsic properties of graph data and their impact on graph learning. In this paper, we narrow the gap by studying the intrinsic dimension of graphs with *Local Intrinsic Dimensionality (LID)*. The LID of a graph measures the expansion rate of the graph as the local neighborhood size of the nodes grows. With LID, we estimate and analyze the intrinsic dimensions of node features, graph structure and representations learned by GNNs. We first show that feature LID (FLID) and structure LID (SLID) are well correlated with the complexity of synthetic graphs. Following this, we conduct a comprehensive analysis of 12 popular graph datasets of diverse categories and show that 1) graphs of lower FLIDs are generally easier to learn; 2) GNNs learn by mapping graphs (feature and structure together) to low-dimensional manifolds that are of much lower representation LIDs (RLIDs), i.e., RLID $\ll$ FLID/SLID; and 3) when the layers go deep in message-passing based GNNs, the underlying graph will converge to a complete graph of SLID $= 0.5$, losing structural information and causing the over-smoothing problem. Finally, we take RLID as an example and showcase that a dimensionality regularizer can help improve existing GNN models.

## 1 INTRODUCTION

Graphs are widely used to model real-life problems owing to their flexible structure and ability to carry different types of information. Graph learning has thus become essential for a wide range of applications in biomedicine (Zitnik et al., 2018), physics (Battaglia et al., 2016) and traffic network (Yu et al., 2018). Whilst the rise of Graph Neural Networks (GNNs) has enabled important breakthroughs in graph learning (Senior et al., 2020; Ying et al., 2018), there is still a lack of understandings of the intrinsic properties of graphs and their impact on learning. In this paper, we narrow this gap by characterizing and analyzing the intrinsic dimensionality (ID) of graphs and graph representations based on an expansion-based intrinsic dimensionality measure: Local Intrinsic Dimensionality (LID). Such an analysis is beneficial for the community to better understand the intrinsic difficulty of a graph learning task and motivate advanced GNNs and learning methods.

The intrinsic dimensionality of a dataset measures the dimension of its underlying manifold or the minimum number of parameters needed to represent the intrinsic structure of the data (Bennett, 1969; Nakada & Imaizumi, 2020). According to the manifold hypothesis (Fefferman et al., 2016) in machine learning, the intrinsic dimensionality is often much lower than the representation dimensionality (the number of features) for real-world high-dimensional data (Tenenbaum et al., 2000; Fodor, 2002; Cayton, 2005; Lin et al., 2006). LID is an expansion-based ID measure associated with the local neighborhood of data points. In other words, the LID of a point measures the intrinsic dimensionality of the local submanifold surrounding the point and the average LID over all points in a set depicts the dimensionality of the entire manifold. The LID metric has been applied to study the intrinsic complexity of many forms of data, such as images, texts and tabular data (Pope et al., 2020; Aghajanyan et al., 2020; Ansuini et al., 2019), as well as the learning and generalization behaviors of deep neural networks. For instance, it has been shown that the LID characteristic of image datasets is closely related to the learning difficulty and generalization performance (Pope et al., 2020).

For graph learning, we are interested in the intrinsic dimensionality of node features, graph structure, representations learned by GNNs and its indication of the final performance. To this end, we apply LID on a diverse set of graph datasets and estimate the Feature LID (FLID), Structure LID (SLID)

and Representation LID (RLID) for each node. The three graph LID measures are then averaged over all nodes in the graph to reflect the overall intrinsic dimensionality. FLID has the same interpretation as the LID for non-graph data. The SLID of a graph can be interpreted as the expansion rate of the graph as the local neighborhood size of its nodes grows. Both FLID and SLID characterize the properties of the raw graph. RLID, on the other hand, characterizes the properties of the integrated representation of both the node feature and the graph structure. With the three LID measures, we provide the following key insights:

- FLID and SLID are good indicators of graph complexity relative to node features and graph structures, respectively. This is verified on synthetic graphs generated using singular value decomposition (SVD) and random geometric graph (RGG).

- With FLID, we study 5 categories of 12 popular graph datasets including co-author graphs, co-purchase graphs, webpage graphs, citation graphs and Wikipedia graphs, and show that graphs of low FLIDs are generally easier to learn and different GNNs are likely to achieve higher accuracies in downstream node classification tasks.

- With RLID and 4 representative GNN models, we show that graph learning is a process that maps the node features and graph structure together onto a simpler manifold that is of a much lower RLID. We also showcase that RLID can be leveraged as a regularizer to improve existing GNN models.

- With SLID, we reveal that the underlying graph converges to a complete graph of $SLID = 0.5$ as the layers of message-passing based GNNs go deep, causing the over-smoothing problem.

## 2   RELATED WORK

Intrinsic dimensionality analysis plays an important role in dimensionality reduction (DeMers & Cottrell, 1993), manifold learning (Law & Jain, 2006), classification (Gong et al., 2019), outlier detection (Houle et al., 2018), generative modeling (Li et al., 2019), adversarial example detection (Ma et al., 2018b), and deep learning understanding (Ma et al., 2018b; Ansuini et al., 2019; Pope et al., 2020). The intrinsic dimensionality of a data representation can be estimated either *globally* on the entire dataset via Principal Component Analysis (PCA) (Wold et al., 1987), graph based methods (Costa & Hero, 2003), and fractal models (Camastra & Staiano, 2016) or *locally* around the individual data points via Local Intrinsic Dimensionality (LID) and its variants (Amsaleg et al., 2015; Houle, 2017; Amsaleg et al., 2019). Different from the global ID measures, LID provides a local view of the intrinsic geometry of the data (see formal definitions in Section 3).

LID has been related to the robustness properties of DNNs to adversarial attacks (Amsaleg et al., 2017; Ma et al., 2018a) and noisy labels (Ma et al., 2018b). It has been shown that the subspaces around adversarial examples are of much higher LID than of the normal examples in the deep representation space of DNNs (Ma et al., 2018a). And when there are noisy labels in the training data, DNN learning exhibits two distinctive phases from dimensionality compression to dimensionality expansion and the expansion phase is when the model starts to overfit the noisy labels (Ma et al., 2018b). The LID of the representations learned by DNNs has also been found to be a good indicator of the generalization performance (Ansuini et al., 2019). Both LID and global ID have been applied to characterize the intrinsic dimensionality of image datasets and representations (Gong et al., 2019; Pope et al., 2020). The intrinsic dimensionality of the objective space (defined by the loss function and model parameters) has also been studied in both natural language processing (Aghajanyan et al., 2020) and computer vision (Li et al., 2018a) to help understand the parameterization redundancy in DNNs. These understandings have motivated either model compression techniques (Li et al., 2018a) or new theories (with the intrinsic parameters) for DNNs (Aghajanyan et al., 2020).

The current understandings of graphs are mostly focused on the expressive power of GNNs. For example, GNNs have been shown to have equivalent discriminative power to the Weisfeiler-Lehman graph isomorphism test (Weisfeiler & Leman, 1968). Xu et al. (2019) showed that GNNs are at most as powerful as the 1-WL test in distinguishing graph structures. Geerts et al. (2021) further proved that degree-aware Message Passing Neural Networks (MPNNs) may be one step ahead of the WL algorithm because of the degree information. Balcilar et al. (2021a) proposed a MPNN model which is experimentally as powerful as a 3-WL test. The learning of GNNs has also been investigated from a spectral perspective. Hoang & Maehara (2019) argued that GNNs only work as a low-pass filter, which was then verified in Balcilar et al. (2021b) by reformulating most of existing GNNs into one common framework. Oono & Suzuki (2019) investigated the asymptotic behaviors of GNNs as the

layer size tended to infinity and related the expressive power of GNNs to the topological information in the spectral domain. In this work, we apply LID to explore the intrinsic complexity of graphs and graph representations, and provide a set of new and complementary insights into graph learning.

## 3 LOCAL INTRINSIC DIMENSIONALITY FOR GRAPHS

### 3.1 LOCAL INTRINSIC DIMENSIONALITY

Given a data set $X \subset \mathbb{R}^n$, $X$ is said to have an intrinsic dimension of $m$ if its elements lie entirely, without information loss, within a $m$-dimensional manifold of $\mathbb{R}^n$, where $m < n$ (Fukunaga, 1982). Before introducing LID, we first explain the intuition behind LID based on the expansion-based modeling of dimensionality. Among the family of dimensionality models, the expansion dimension (ED) (Karger & Ruhl, 2002) quantify the ID in the vicinity of a point of interest in the data domain. More precisely, it assesses the rate of growth in the number of data points encountered as the distance from the reference point increases.

As an example, in the Euclidean space $\mathbb{R}^m$, one can measure the volume $V_i$ of a $m$-ball of radius $r_i$ with $i \in \{1, 2\}$, taking the logarithm of the ratio would reveal the dimension $m$: $\frac{V_2}{V_1} = \left(\frac{r_2}{r_1}\right)^m \Rightarrow m = \frac{\ln(V_2/V_1)}{\ln(r_2/r_1)}$. Transferring the concept of expansion dimension to the statistical setting with neighborhood distance distributions gives us the formal definition of LID (Houle, 2017).

**Definition 1 (Local Intrinsic Dimensionality)** *Given a data sample $x \in X$, let $R > 0$ be a random variable denoting the distance from $x$ to other data samples. If the cumulative distribution function $F(r)$ of $R$ is positive and continuously differentiable at distance $r > 0$, the LID of $x$ at distance $r$ is given by:*

$$LID_F(r) \triangleq \lim_{\epsilon \to 0} \frac{F((1 + \epsilon)r) - F(r))}{\epsilon \cdot F(r)} = \frac{r \cdot F^{'}(r)}{F(r)}, \qquad (1)$$

*The local intrinsic dimension at $x$ is then defined as the limit, as the radius $r$ tends to zero, i.e. $LID_F \triangleq \lim_{r \to 0} LID_F(r)$.*

Here, the CDF $F(r)$ is analogous to the volume in the Euclidean example. Since $F(r)$ is unknown, estimators are needed for LID. There already exist a number of LID estimators in the literature (Levina & Bickel, 2005; Amsaleg et al., 2015; Liao et al., 2014). In the following, we will introduce one commonly used LID estimator and how it can be applied on graphs.

### 3.2 FEATURE AND REPRESENTATION LID

For graphs, we are interested in the LIDs of the nodes features and structure of the graph itself and node representations learned by GNNs. Node features and graph structure are two fundamental information of the graph, while the learned representation for a node is an integration of its feature and the structural information. The existing LID estimators developed for non-graph data can be directly applied to node features and node representation. Here, we first introduce the LID estimation for Feature LID (FLID) and Representation LID (RLID).

Amongst the existing LID estimators, the Maximum Likelihood Estimator (MLE) (Levina & Bickel, 2005; Amsaleg et al., 2015) is one of the most cited estimators. It treats the neighbors of each point $x \in X$ as events in a Poisson process and the distance $r^{(j)}(x)$ between $x$ and its $j$-th nearest neighbor as the event's arrival time. Since this process depends on the dimensionality $d$, MLE estimates the intrinsic dimension by maximizing the log-likelihood of the observed process.

The node features or representations are represented as vectors in the Euclidean space. Thus, FLID and RLID can be directly estimated by MLE. Let $x$ denote the feature/representation vector of a particular node, the FLID/RLID of $x$ can be estimated as following:

$$\text{FLID/RLID}(x, k) = \left(\frac{1}{k} \sum_{j=1}^{k} log \frac{r^{(k+1)}(x)}{r^{(j)}(x)}\right)^{-1}, \qquad (2)$$

where $k$ is the neighborhood size (i.e., $k$-nearest) and $r^{(i)}(x)$ is the Euclidean distance between $x$ and its $i$-th nearest neighbor. Averaging $\text{FLID}(x, k)$ across all nodes $x$ in $\{x_i\}_{i=1}^{N}$ leads to

the FLID of the entire graph, i.e., $\text{FLID}_G(k) = \frac{1}{N} \sum_{i=1}^{N} \text{FLID}(\boldsymbol{x}_i, k)$. Similarly, we can obtain $\text{RLID}_G(k) = \frac{1}{N} \sum_{i=1}^{N} \text{RLID}(\boldsymbol{x}_i, k)$. The $k$-nearest neighbors are identified based on the pairwise distance between all nodes in the graph.

### 3.3 STRUCTURAL LID

For graph structure, the distance between a pair of nodes is an integer $y \in [1, 2, \dots]$, namely hops. In Definition 1, LID is defined as the normalized rate of increase of the neighborhood size. However, the neighborhood size in a graph often grows exponentially as the hop increases, i.e., $G(y) = a^y$ at hop $y$ Ritter et al. (2018). Intuitively, the base that best characterizes the growth rate of the entire graph is its average degree. Locally, different nodes should have different bases $a \in \mathbb{R}^+$ that best characterize their own neighborhoods. With exponentially growing neighborhood, the LID of a node becomes zero:

$$LID_G = \lim_{y \to 0} LID_G(y) = \lim_{y \to 0} \frac{y \cdot \frac{\mathrm{d}}{\mathrm{d}y} a^y}{a^y} = \lim_{y \to 0} y \cdot \ln a = 0. \tag{3}$$

To solve this issue, Ritter et al. (2018) assume that $G(y)$ is created by applying a logarithmic transformation to a new distance variable $r$ of $F(r)$ ($y \triangleq \ln r$), leading to the definition of the *intrinsic degree* $\log LIB_G(y) \triangleq LID_F(r)$. This relates variable $y$ to a new variable $r$ whose LID can be estimated by the MLE (Amsaleg et al., 2015). Relating back to variable $y$ with the estimated $LID_F(r)$ gives us the SLID of the graph:

$$\text{SLID}(x, r) = \left( r - \frac{1}{k} \sum_{i=1}^{k} y_i \right)^{-1}, \quad \text{SLID}_G(r) = \frac{1}{N} \sum_{i=1}^{N} \text{SLID}(x_i, r), \tag{4}$$

where $x$ is a node, $y_i$ is the $i$-nearest neighbor distance and $r = \max\{y_i, \dots, y_k\}$ is the neighborhood radius. Note that the neighbors are automatically determined by $r$.

## 4 VALIDATING FLID AND SLID ON SYNTHETIC GRAPHS

In the dimensionality estimation literature, synthetic data with fixed intrinsic dimensionalities are often used to test the accuracy of the estimation methods. However, graphs are complex data whose intrinsic dimensionality is hard to simulate. To overcome this issue, here we use Singular Value Decomposition (SVD) applied on the feature matrix of an existing graph to generate synthetic graphs of varying feature complexities. For structural analysis, we use a traditional graph generation model Random Geometric Graph (RGG) to generate synthetic graphs of varying structural complexities.

### 4.1 VALIDATING FLID

SVD is a classic dimensionality reduction method. A matrix $\boldsymbol{X} \in \mathbb{R}^{m \times n}$ can be factorized into the product of three matrices $\boldsymbol{X} = \boldsymbol{U} \boldsymbol{\Sigma} \boldsymbol{V}^T$, where $\boldsymbol{U} \in \mathbb{R}^{m \times m}$ and $\boldsymbol{V} \in \mathbb{R}^{n \times n}$ are both a unitary matrix and $\boldsymbol{\Sigma} \in \mathbb{R}^{m \times n}$ is a diagonal matrix. Let $\boldsymbol{u}$ and $\boldsymbol{v}$ be the column vectors of $\boldsymbol{U}$ and $\boldsymbol{V}$ respectively, the vector form of the decomposition is written as $\boldsymbol{X} = \sum_{i=1}^{r} \sigma_i \boldsymbol{u}_i \boldsymbol{v}_i^T$, where $\sigma$ is the singular value. For $s \in \{1, \dots r\}$, extracting the top $s$ singular values and their corresponding vectors results in a truncated summation $\boldsymbol{X}_k = \sum_{i=1}^{s} \sigma_i \boldsymbol{u}_i \boldsymbol{v}_i^T$, which is proved to be the best rank $k$ approximation to $\boldsymbol{X}$ in both Frobenius norm and $L_2$-norm (Horn & Johnson, 1985).

To test FLID, we choose three popular citation datasets Cora, CiteSeer and PubMed (Sen et al., 2008; Namata et al., 2012) to construct synthetic graphs using the above method. The feature vectors of each dataset are decomposed into the form $\boldsymbol{X} = \sum_{i=1}^{s} \sigma_i \boldsymbol{u}_i \boldsymbol{v}_i^T$. Varying the value of $s$, we generate a set of synthetic vectors with different ranks. The synthesized data shows growing feature complexity as the rank increases. Figure 1 visualizes the t-SNE (Van der Maaten & Hinton, 2008) 2-D embedding of the generated features from Cora. The graph structure remains the same.

Following Equation (2), We estimate FLID for each set of synthetic vectors and plot the trend with increasing $s$ in Figure 2. It shows that the FLID of the synthetic data increases with $s$. This verifies that the intrinsic dimensionality increases as we keep more information of the original features, which matches our expectation. Although the estimation is sensitive to the choice of neighborhood size $k$ to some extent, the trend is consistent across different $k$ values. The different growing speed of FLID on different graphs indicates their differences in the feature distribution at different ranks.

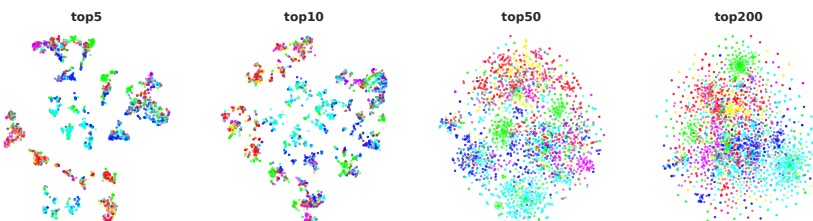

**Figure 1.** The t-SNE visualization of the synthesized feature vectors using SVD from Cora dataset. Each class label is assigned to a unique color. We extract the top $s = 5, 10, 50, 200$ singular terms of the original features and construct a $s$-rank approximation.

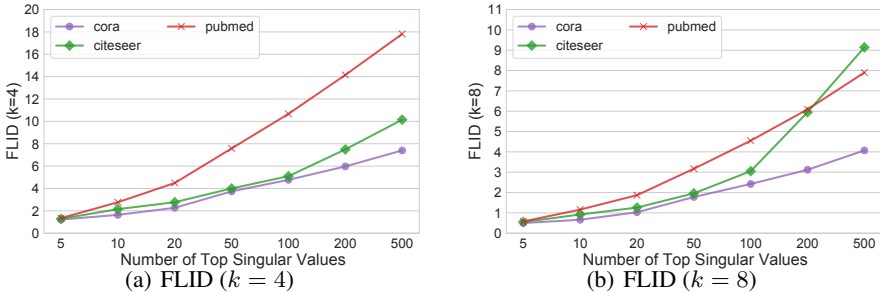

**Figure 2.** FLID values of synthetic data generated from Cora, CiteSeer and PubMed. For each set of synthetic data, FLID increases with the growing number of top singular values. FLID is estimated with two different neighborhood sizes: $k = 4$ (**left**) and $k = 8$ (**right**).

## 4.2 VALIDATING SLID

In graph theory, a random geometric graph (RGG) (Gilbert, 1961) is a generative latent point model where each node is assumed to be associated with a latent point in a metric space $\mathbb{R}^d$. Let $G = (V, E)$ denote an undirected graph with a vertex-set $V$ and a edge-set $E$. Considering a metric space $[0, 1)^d$ with Euclidean distance, it first randomly samples $n = |V|$ independent and identically distributed latent points $\{\boldsymbol{x}_i\}_{i=1}^n$ from the underlying space. Two vertices $p, q \in V$ are connected if and only if the distance $\|\boldsymbol{x}_p - \boldsymbol{x}_q\|$ is less than a specified threshold $\tau$. Thus, the parameters $n$ and $\tau$ fully characterize an RGG.

When changing the dimension $d$ of the latent space, the profile of the generated topology structure also changes, as shown in Figure 3. Intuitively, the (intrinsic) dimensionality of the generated graph should be higher if the dimension of the latent space increases. To generate more diverse RGGs, here we also varies the connection probability $p$ and the vertex set size $n$. As suggested in (Dall & Christensen, 2002), the threshold $\tau$ can be calculated by $\tau = \frac{1}{\sqrt{\pi}} \left[ p\Gamma(\frac{d+2}{2}) \right]^{\frac{1}{d}}$, where $\Gamma$ is the gamma function. As shown in Figure 4, in general, SLID is positively correlated with the latent space dimensionality of the synthetic graphs. In general, SLID is positively correlated with the latent space dimensionality of the synthetic graphs. One exception (the two points in the red circle) occurs when estimating SLID using radius $r = 4$. We conjecture this slight variation is caused by the estimation instability at a large radius (i.e., $r = 4$ vs. $r = 3$). A similar sensitivity has also been observed on image datasets in (Pope et al., 2020). As such, we will use $r = 3$ for SLID estimation in the rest of our experiments. It is also worth mentioning that SLID increases slowly but almost linearly with latent space dimensionality for $d \leq 10$; it then grows drastically for $d > 10$. This indicates that synthesized graphs of $d > 10$ have structural properties that are significantly different (intrinsically more complex) from those of $d \leq 10$, as can be visually verified in Figure 3.

## 5 CHARACTERIZING REAL-WORLD GRAPHS WITH FLID AND SLID

In this section, we estimate the FLID and SLID of 12 popular graph datasets and show that real-world graphs are of much lower feature and structure intrinsic dimensionalities relative to their high extrinsic dimensions (nodes and edges). To ensure the generality of our findings, here we consider 5 different categories of graph datasets: Citation, Wikipedia, WebKB, Co-author and Co-purchase.

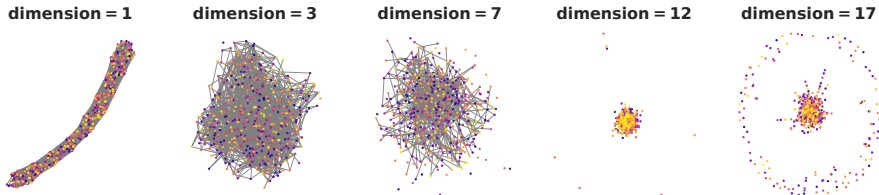

**Figure 3.** The visualization of synthetic RGG graphs with varying latent space dimensions $d = 1, 3, 7, 12, 17$. The set of graphs show various profiles as the latent space dimensionality changes.

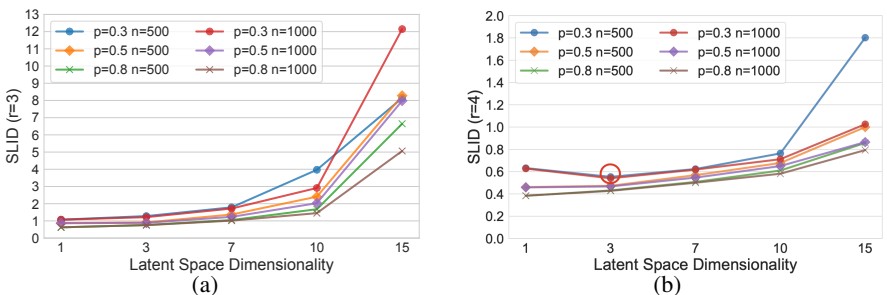

**Figure 4.** The trend of SLID with increasing latent space dimensionality $d$. We vary the connection probability $p$ in the range $[0.3, 0.5, 0.8]$ and the vertex set size $n$ in the range $[500, 1000]$. To see the sensitivity of the SLID estimation to neighborhood radius $r$, here we test two radii $r = 3, 4$.

**Citation Network** Cora, CiteSeer and PubMed are three popular citation graph datasets (Sen et al., 2008; Namata et al., 2012). In these graphs, nodes represent papers and edges correspond to the citation relationship between two papers. We transform the node features of PubMed to the bag-of-words representation of papers, keeping in line with Cora and CiteSeer. Nodes are classified according to academic topics.

**Wikipedia Network** Chameleon and Squirrel (Rozemberczki et al., 2021) are Wikipedia page networks on specific topics, where nodes represent web pages and edges are the mutual links between them. Node features are the bag-of-words representation of informative nouns. The nodes are classified into four categories according to the number of the average monthly traffic of the page.

**WebKB** Cornell, Wisconsin and Texas (Craven et al., 1998) are three subsets of WebKB, a webpage dataset collected from computer science departments of three universities. Nodes are web pages and edges are hyperlinks between them. Node features are the bag-of-words representation of web pages. Each node is labeled to be a 'student', 'project', 'course', 'staff' or 'faculty'.

**Co-author Network** Cs and Physics are co-author networks constructed from Microsoft Academic Graph (McAuley et al., 2015; Shchur et al., 2018). Nodes denote authors and edges indicate whether two authors are co-authors in a paper. Node features are paper keywords extracted from papers authored by a particular author. An author's most active field of study is used as the node label.

**Amazon Co-purchase Network** Photo and Computers (Shchur et al., 2018) were collected by crawling Amazon websites. Goods are represented as nodes and the co-purchase relationships are denoted as edges. Node features are the bag-of-words representation of product reviews. Each node is labeled with the category of goods.

The FLID and SLID values of the 12 datasets are shown in Table 1. We find that, from the FLID perspective, the intrinsic dimensionality ranking is: Co-purchase graphs ¿ Wikipedia graphs ¿ Citation graphs ¿ WebKB graphs ¿ Co-author graphs. This indicates that the node features of one category of graphs share intrinsic properties that may be very different from that of other categories. This is not a surprise since the most discriminative node attributes of different types of graphs are indeed different. Interestingly, graphs from different categories can also have similar intrinsic dimensional properties, e.g., Cora has more similar FLID to Cornell/Texas than to CiteSeer or PubMed. From the SLID perspective, there is no consistent relationship between two graph categories. It is worth mentioning that the SLID property is rather dataset-dependent, as we have shown in Figure 4 that the same type of graphs can have very different SLIDs. From a node-level view, the number of features and 3-hop degree (the same neighbourhood as we use to calculate SLID) can be regarded as *extrinsic* feature and structural dimensionalities of a given node, respectively. Comparing the extrinsic

**Table 1.** The FLID and SLID of 12 real-world graph datasets. For estimation, $k = 4$ is used for FLID while $r = 3$ is used for SLID. Other graph attributes are also reported, including the number of nodes, edges, node features, average degree and 3-hop degree (number of 3-hop neighbors).

| Category | Dataset | FLID | SLID | #Nodes | #Edges | #Features | Degree | 3-Hop Degree |
|---|---|---|---|---|---|---|---|---|
| | Cora | 9.10 | 10.20 | 2708 | 5278 | 1433 | 4.90 | 128.08 |
| Citation | CiteSeer | 18.18 | 3.78 | 3327 | 4552 | 3703 | 3.77 | 43.51 |
| | PubMed | 17.81 | 37.09 | 19717 | 44324 | 500 | 5.50 | 394.62 |
| Wikipedia | Chameleon | 26.46 | 8.46 | 2277 | 36101 | 500 | 5.0 | 1067.34 |
| | Squirrel | 51.57 | 14.19 | 5201 | 217073 | 2089 | 154.0 | 3639.55 |
| | Cornell | 8.66 | 6.56 | 183 | 295 | 1703 | 1.0 | 116.04 |
| WebKB | Wisconsin | 5.76 | 7.41 | 251 | 309 | 1703 | 6.0 | 158.31 |
| | Texas | 8.66 | 6.19 | 183 | 499 | 1703 | 2.0 | 127.83 |
| Co-author | Cs | 4.80 | 42.37 | 18333 | 81894 | 500 | 8.93 | 873.62 |
| | Physics | 2.92 | 57.09 | 34493 | 247962 | 500 | 14.38 | 2428.83 |
| Co-purchase | Photo | 74.26 | 10.90 | 7650 | 119081 | 745 | 18.78 | 649.16 |
| | Computers | 80.29 | 20.46 | 13752 | 245861 | 767 | 20.88 | 1365.58 |

dimensionalities with the intrinsic dimensionalities FLID and SLID, we find that real-world graphs actually have much lower intrinsic dimensionalities, indicating that there exist a low-dimensional re-parameterization for complex graph data.

# 6 UNDERSTANDING THE LEARNING PROCESS OF GNNS

In this section, we apply FLID, RLID and SLID to understand the learning process of GNNs. Note that most graph learning methods do not change the graph's topology, i.e., *SLID stays the same before, during or after learning*. FLID measures the property of the raw graph before learning while RLID measures the property of the learned representation during or after learning. We consider four GNN models of different design principles: GCN (Kipf & Welling, 2016), GAT (Veličković et al., 2018), GCNII (Chen et al., 2020) and SplineCNN (Fey et al., 2018). All models are implemented with Pytorch Geometric (PyG) (Fey & Lenssen, 2019). The optimal parameter settings suggested by the original papers are used. For datasets that were never reported with a particular GNN, we perform a hyper-parameter search on the validation set for dropout rate, learning rate, and the number of hidden units. Each dataset is divided into proportions 60%, 20% and 20% for training, validation and testing, respectively. The models are tested on the node classification task.

## 6.1 GRAPHS WITH LOWER FLIDS ARE EASIER TO LEARN

The performance of different GNN models are shown in Table 2. An interesting observation is that the relative ranking of the test accuracy between the 4 models are almost the same across the same category of datasets. This indicates that the models' performance may be determined by certain inherent properties of the graphs. We then investigate this conjecture with FLID and other graph properties, including the number of nodes, the number of edges, and average degree. Among these properties, only FLID demonstrates a strong relationship with the test accuracy. As shown in Figure 5, within each category, GNNs generally have better performance on datasets of lower FLIDs. This implies that graphs of low FLIDs are generally easier to learn, regardless of the GNN model used for learning. It also indicates that node features may play a key role in determining the overall learning complexity of the graph. By contrast, the performance has no consistent correlation with other graph properties including the number of nodes, number of edges, average degree and SLID, as shown in Appendix B. The SLID is not a good indicator of the final performance is because *the structure of the graph does not change before, during, or after learning*, and arguably, the static structure itself is not sufficient to indicate learning difficulty.

## 6.2 GNNS LEARN LOW-DIMENSIONAL REPRESENTATION SPACE

Here, we delve into the learning process of GNNs and show that GNNs learn by mapping the original data onto a simpler manifold in the presentation space that is of much lower intrinsic dimensionality. Here, we are interested in the FLID of raw node features and the RLID of learned representations. At each training epoch, we extract the outputs of the last hidden layer of the GNN model as the representations and estimate its RLID following Equation (2). Figure 6 illustrates the across-epoch changes of RLID and FLID on datasets Cora, CiteSeer and Pubmed. The results of other datasets

**Table 2.** Performance (test accuracy in node classification) of 4 GNN models on 12 graph datasets. For all models, the accuracy ranking across the datasets are almost the same: CiteSeer < PubMed < Cora; Squirrel < Chameleon; Cornell < Wisconsin < Texas; Cs < Physics; Computers < Photo.

| Category | Dataset | FLID | Test Accuracy(%) | | | |
|---|---|---|---|---|---|---|
| | | | GCN | GAT | GCNII | SplineCNN |
| Citation | Cora | 9.10 | 85.06 | 88.75 | 91.33 | 91.05 |
| | CiteSeer | 18.18 | 75.08 | 77.78 | 78.98 | 79.40 |
| | PubMed | 17.81 | 87.60 | 85.37 | 87.83 | 89.65 |
| Wikipedia | Chameleon | 26.46 | 39.69 | 44.74 | 42.54 | 46.41 |
| | Squirrel | 51.57 | 29.68 | 30.45 | 31.22 | 40.87 |
| WebKB | Cornell | 8.66 | 56.78 | 62.16 | 64.86 | 77.78 |
| | Wisconsin | 5.76 | 58.82 | 62.75 | 64.71 | 86.11 |
| | Texas | 8.66 | 67.57 | 70.27 | 72.97 | 81.48 |
| Co-author | Cs | 4.80 | 92.88 | 89.5 | 91.57 | 93.72 |
| | Physics | 2.92 | 95.70 | 91.2 | 96.01 | 97.41 |
| Co-purchase | Photo | 74.26 | 91.31 | 90.26 | 90.72 | 95.75 |
| | Computers | 80.29 | 80.81 | 72.34 | 75.03 | 88.95 |

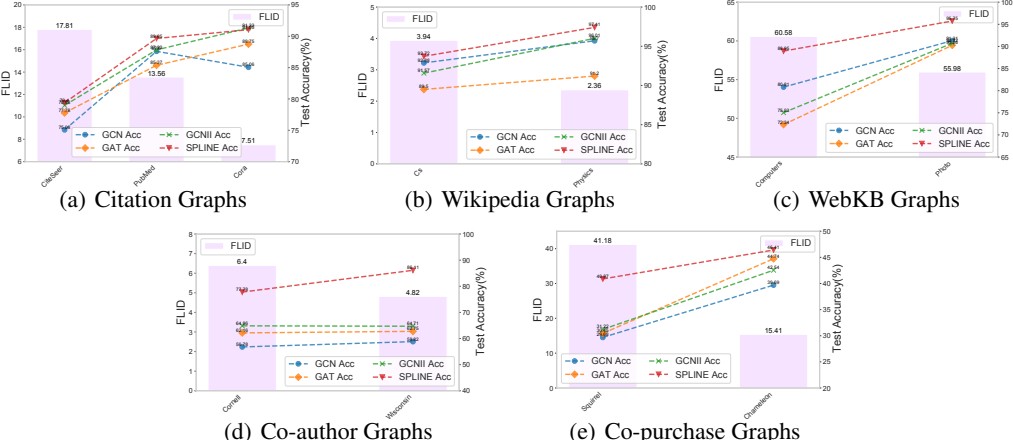

(a) Citation Graphs     (b) Wikipedia Graphs     (c) WebKB Graphs

(d) Co-author Graphs     (e) Co-purchase Graphs

**Figure 5.** Correlation Between Test Accuracy and FLID

are shown in Appendix C. One key observation is that the FLID is significantly reduced once the learning starts. This indicates that RLID is much lower than FLID, that is, the representation space is much simpler (or of much lower intrinsic dimensionality) than the raw feature space. It also suggests that the GNN learning actually drives down the intrinsic dimensionality of datasets, potentially changing a messy node distribution to a simpler profile. Moreover, the RLIDs of the 4 models converge to a similar value at the last few epochs. It reveals some consistency across different GNN models of diverse frameworks and theoretical bases.

One might expect that RLID should have a simple relationship with FLID and SLID, as graph representation learning is a fusion of both feature and structural information. However, here the three LID measures cannot seem to reveal this relationship. This because graph learning is a complex process that involves the transformation of the underlying k-nearest neighbor graph (k-NNG) of the original graph to a new k-NNG in the representation space, i.e., neighbor nodes in the original graph should have similar representation vectors. And during this process, the original graph structure is static, i.e., *SLID stays the same before, during, or after learning*. So, it is hard to study the structural transformation using SLID of the original graph. Interestingly, a recent work also shows that the structure of the learned representation is very different from the original structure using constructed k-NNGs from the original graph and learned representations (Jin et al., 2021). We will study the SLID of the evolving underlying k-NNG in our future work.

### 6.3 DROPEDGE BRINGS DOWN SLID

For most GNN models, the graph topology remains unchanged during training. Therefore, it is hard to interpret the learning process from the structure perspective. Here, we turn to investigate the

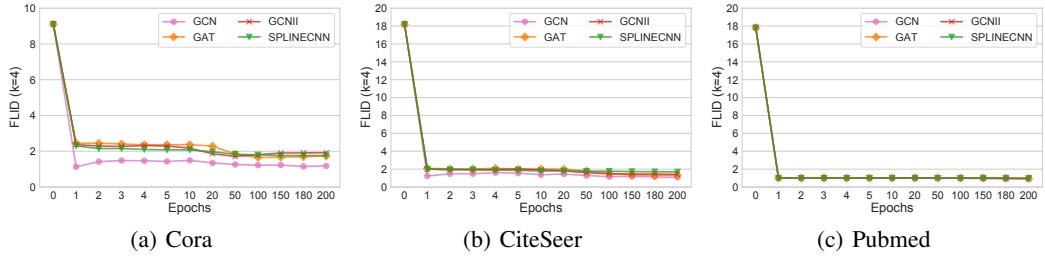

**Figure 6.** The change of FLID (epoch 0) and RLID (epoch 1-200) across different epochs. FLID is estimated on the raw node features, while RLID is estimated on the outputs of the 4 models on Cora, CiteSeer and Pubmed. The neighborhood size $k = 4$ is used for estimating both FLID and RLID.

impact of DropEdge (Rong et al., 2020) to the structural property. DropEdge randomly removes a certain number of edges from the input graph at each training iteration, which has led to improved performance on a variety of GNNs. It has been explained that DropEdge works either by retarding the convergence speed of over-smoothing or relieving the information loss (Rong et al., 2020).

Here, we empirically show that DropEdge indeed reduces the SLID of the graph. We estimate the SLIDs of the new graphs generated by DropEdge at various dropping rates and the model's performance on the new graphs. The results on Core, CiteSeer and PubMed are shown in Figure 7. It is clear that SLID is reduced linearly as the dropping rate increases. It also shows that the test accuracy tends to rise until more than 50% of the edges are dropped. This is not so surprising as losing too much of the intrinsic structure will destroy the topology of the graph. From the perspective of intrinsic dimensionality, dropping edge is a dimensionality reduction technique that reduces the manifold complexity and in turn the learning difficulty, if not causing too much information loss.

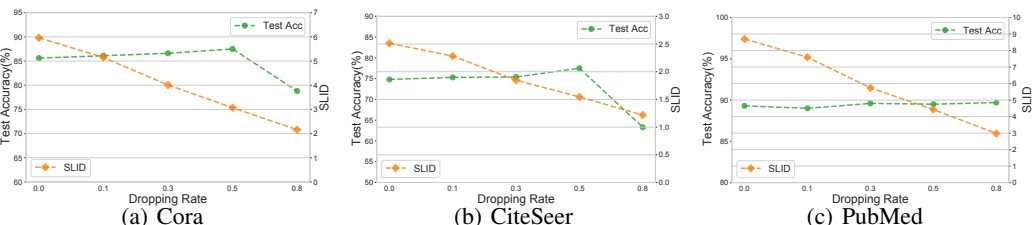

**Figure 7.** The SLID and test accuracy at different dropping edges ($p \in [0, 0.8]$) on Cora, CiteSeer and PubMed. $p = 0$ marks the raw graphs.

### 6.4 UNDERSTANDING OVER-SMOOTHING WITH SLID

The over-smoothing problem of GNNs refers to the representation collapse when stacking more layers in GNNs, causing a significant performance drop (Li et al., 2018b). Here, we provide a new explanation for the over-smoothing problem using SLID. According to (Balcilar et al., 2021b), most of the well-known GNN models can be re-formulated into a general form: $H^{l+1} = \sigma\left(\sum_s C^{(s)} H^{(l)} W^{(l,s)}\right)$, where $C^{(s)}$ is the $s$-th convolution support that defines how the node features are propagated to the neighbors, and $H^{(l)}$ and $W^{(l,s)}$ are the hidden representation and weights for $s$-th convolution support, respectively. Here, we focus on a particular class of GNNs whose convolution support can be represented by the adjacency matrix $A$. Representative models from this class include GCN with $C = D^{-\frac{1}{2}} A D^{-\frac{1}{2}}$ ($D$ is the degree matrix) and GIN (Balcilar et al., 2021b) with $C = A + (1 + \epsilon)I$. For these models, $A$ is multiplied cumulatively with more layers, resulting in a dynamic graph with new edges. For example, given a graph with adjacency matrix $A$, connecting the nodes that share a common neighbor yields a new graph with $A^2$.

By analyzing the SLID of the accumulated adjacency matrix, we are able to reveal the structural collapse of the underlying graph when stacking more layers. The SLIDs of accumulated matrices $A, A^2, \ldots, A^6$ are reported in Table 3, where it shows that the SLID of all 12 graph datasets converges to $SLID = 0.5$ after the 4th iteration. This indicates that the underlying graph converges to a complete graph when more layers are added to the model, losing the original topology. In a complete graph, each pair of distinct vertices has a one-hop neighborship. In Equation 4, letting $r = 3$, a complete graph means $k = n$ and $y_1 = \cdots = y_n = 1$, leading to SLID = $\left(3 - (1/n) \cdot n\right)^{-1} = 0.5$. This provides a new perspective of explanation for the over-smoothing problem of GNNs.

**Table 3.** The SLID of a set of adjacency matrices generated via cumulative multiplication. The radius $r = 3$ is used for the SLID estimator.

| Dataset | $A$ | $A^2$ | $A^3$ | $A^4$ | $A^5$ | $A^6$ |
|---------|-----|-------|-------|-------|-------|-------|
| Cora | 10.204 | 9.514 | 1.106 | 0.528 | 0.495 | 0.495 |
| CiteSeer | 3.783 | 5.762 | 2.643 | 0.630 | 0.477 | 0.473 |
| PubMed | 37.091 | 37.370 | 1.043 | 0.517 | 0.500 | 0.500 |
| Chameleon | 8.460 | 1.508 | 0.561 | 0.500 | 0.500 | 0.500 |
| Squirrel | 14.192 | 1.027 | 0.526 | 0.500 | 0.500 | 0.500 |
| Cornell | 6.559 | 0.975 | 0.533 | 0.499 | 0.499 | 0.499 |
| Wisconsin | 7.410 | 1.078 | 0.539 | 0.499 | 0.499 | 0.499 |
| Texas | 6.193 | 0.889 | 0.523 | 0.499 | 0.499 | 0.499 |
| Cs | 42.374 | 13.848 | 0.934 | 0.506 | 0.500 | 0.500 |
| Physics | 57.085 | 13.268 | 0.802 | 0.501 | 0.500 | 0.500 |
| Photo | 3.915 | 2.674 | 1.037 | 0.560 | 0.491 | 0.491 |
| Computers | 20.459 | 3.733 | 0.801 | 0.518 | 0.497 | 0.497 |

**Table 4.** Test accuracy (%) of the GCN (Kipf & Welling, 2016) model trained with or without RLID regularization.

| Loss$\downarrow$, Dataset$\rightarrow$ | Cora | Citeseer | Pubmed |
|---|---|---|---|
| CE (Kipf & Welling, 2016) | 84.7 | 75.4 | 88.6 |
| $\ell_{\mathrm{RLID}}$ ($\lambda = 1$) | **85.3** | **75.8** | **89.0** |

# 7 IMPROVING PERFORMANCE OF GNNS USING RLID

The three LID metrics can be utilized as regularizers or supervision signals to guide graph learning towards more locally discriminable (low LIDs) representations. Here, we take RLID as an example to regularize the model ($f_\theta$) to learn low-dimensional representations. The regularized objective can be written as:

$$\ell_{\mathrm{RLID}}(\boldsymbol{x}) = \ell(f_\theta(\boldsymbol{x}), y) + \lambda \cdot RLID(f_\theta(\boldsymbol{x})), \tag{5}$$

where $\ell$ denotes the commonly used cross-entropy (CE) loss and $\lambda$ is the coefficient of the RLID regularization term.

We empirically evaluate this regularized objective on Cora, Citeseer, and Pubmed, and show that it can improve existing models. As shown in Table 4, it improves the performance of GCN by 0.6% on Cora, 0.4% on Citeseer, and 0.4% on Pubmed. Note that this is just a simple attempt and the improvement is not so significant. We believe that a more advanced strategy is to exploit the FLID or SLID of each node as *targets* of the dimensionality regularizer to prevent the collapse of the intrinsic structure in the presentation space and thus the over-smoothing problem. We will leave these explorations to our future work.

# 8 CONCLUSION

In this work, we investigated the intrinsic dimensionality of node features, graph structure and representations learned by GNNs with Local Intrinsic Dimensionality (LID). Estimators for Feature LID (FLID), Structure LID (SLID) and Representation LID (RLID) were introduced and verified on synthetic graphs. With FLID and SLID, we showed that real-world graphs have much lower intrinsic dimensionalities than their extrinsic dimensionalities. With FLID and RLID, we revealed that GNNs learn to map the raw features and structure to a representation space that is of much lower intrinsic dimensionality. With SLID, we found that DropEdge not only removes edges but also reduces the complexity of the intrinsic structure and the learning difficulty, and that over-smoothing is caused by the collapse of the graph structure to a complete graph of $SLID = 0.5$. These understandings could help motivate more advanced graph learning techniques.

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

## A  SLID AND FLID OF GRAPH DATASETS

Table 5 and Table 6 detail the FLID of SLID of all 12 datasets with various parameter settings. For FLID, the nearest neighborhood size $k$ in Equation 2 is tuned from 2 to 8, following the setting in Ma et al. (2018a). For SLID, the radius $r$ in Equation 4 changes in the range 2 to 5, as indicated in Ritter et al. (2018).

**Table 5.** FLID of Real-world Graph Datasets

| Category | Dataset | k=2 | k=3 | k=4 | k=5 | k=6 | k=7 | k=8 |
|---|---|---|---|---|---|---|---|---|
| | Cora | 19.40 | 11.89 | 9.10 | 7.51 | 6.44 | 5.65 | 5.04 |
| Citation | CiteSeer | 22.69 | 18.86 | 18.18 | 17.81 | 17.70 | 17.67 | 17.49 |
| | PubMed | 49.28 | 26.01 | 17.81 | 13.56 | 10.97 | 9.19 | 7.90 |
| Wikipedia | Chameleon | 65.41 | 33.90 | 26.46 | 15.41 | 14.00 | 12.24 | 10.82 |
| | Squirrel | 165.53 | 73.01 | 51.57 | 41.18 | 34.22 | 30.43 | 27.14 |
| | Cornell | 26.58 | 13.18 | 8.66 | 6.40 | 5.20 | 4.48 | 3.92 |
| WebKB | Wisconsin | 14.38 | 7.28 | 5.76 | 4.82 | 4.10 | 3.57 | 3.19 |
| | Texas | 26.58 | 13.18 | 8.66 | 6.40 | 5.20 | 4.48 | 3.92 |
| Co-author | Cs | 11.24 | 6.40 | 4.80 | 3.94 | 3.35 | 2.96 | 2.64 |
| | Physics | 7.38 | 4.02 | 2.92 | 2.36 | 2.02 | 1.78 | 1.60 |
| Co-purchase | photo | 232.42 | 112.72 | 74.36 | 55.98 | 44.95 | 35.45 | 30.23 |
| | Computers | 243.26 | 121.37 | 80.29 | 60.58 | 48.67 | 40.62 | 34.81 |

**Table 6.** SLID of Real-world Graph Datasets

| Category | Dataset | k=2 | k=3 | k=4 | k=5 |
|---|---|---|---|---|---|
| | Cora | 5.96 | 10.20 | 9.14 | 1.07 |
| Citation | CiteSeer | 2.51 | 3.78 | 5.37 | 2.45 |
| | PubMed | 8.70 | 37.09 | 29.80 | 1.04 |
| Wikipedia | Chameleon | 44.90 | 8.46 | 1.47 | 0.50 |
| | Squirrel | 60.71 | 14.19 | 1.02 | 0.46 |
| | Cornell | 13.89 | 6.56 | 0.95 | 0.47 |
| WebKB | Wisconsin | 14.81 | 7.41 | 1.05 | 0.48 |
| | Texas | 15.86 | 6.19 | 0.87 | 0.45 |
| Co-author | Cs | 8.70 | 42.37 | 13.47 | 0.92 |
| | Physics | 12.17 | 57.09 | 12.96 | 0.80 |
| Co-purchase | Photo | 8.06 | 10.90 | 4.70 | 1.06 |
| | Computers | 11.62 | 20.46 | 3.67 | 0.78 |

# B CORRELATIONS BETWEEN MODEL PERFORMANCE AND COMMON GRAPH PROPERTIES

Figure 8, 9, 10 and 11 show the correlation between test accuracy and graph properties including the number of nodes, the number of edges, the average degree and SLID. As can be observed, the first three properties demonstrate inconsistent correlations across different datasets.

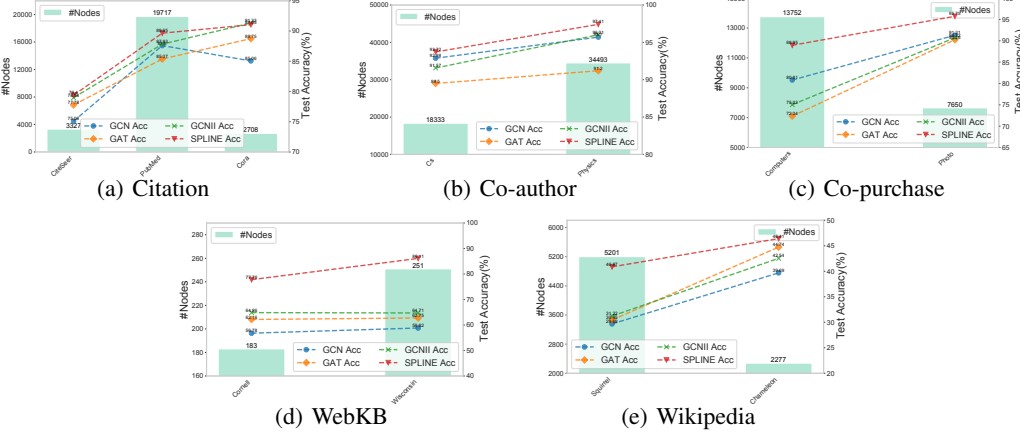

**Figure 8.** Correlation Between Test Accuracy and Number of Nodes

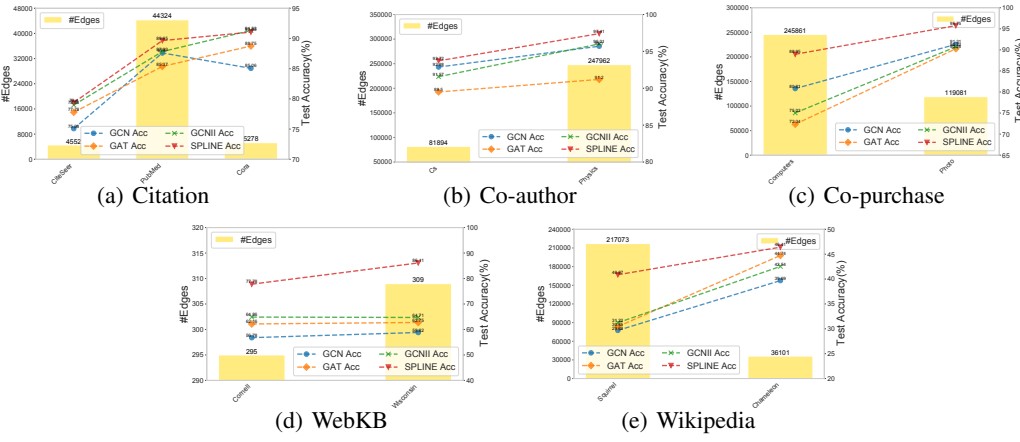

**Figure 9.** Correlation Between Test Accuracy and Number of Edges

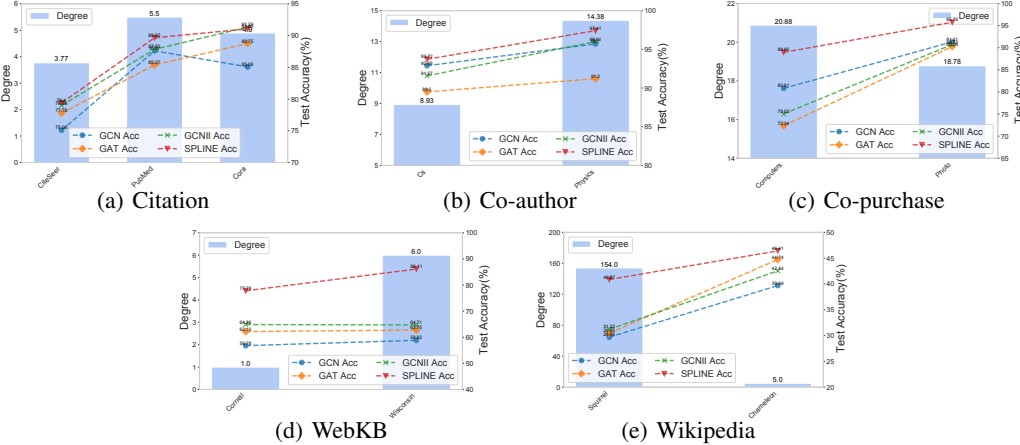

**Figure 10.** Correlation Between Test Accuracy and Average Degree

## C    CHANGES OF RLID ACROSS EPOCH AND FLID OF DATASETS

Figure 12 shows the FLID and across-epoch RLID on the rest of the 9 graph datasets. It can be observed that RLID is significantly lower than FLID on all the datasets, and RLIDs of the four GNN models converge to a similar value at the last few epochs. Note that FLID is computed on the raw node features before training and RLID is computed at each training epoch. Therefore, FLID only appears at the first point (epoch 0) in each plot.

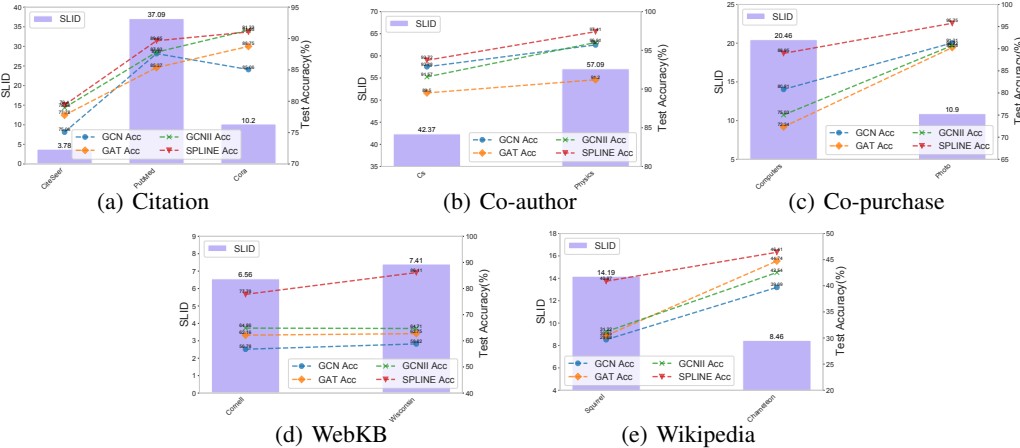

**Figure 11.** Correlation between Test Accuracy and SLID

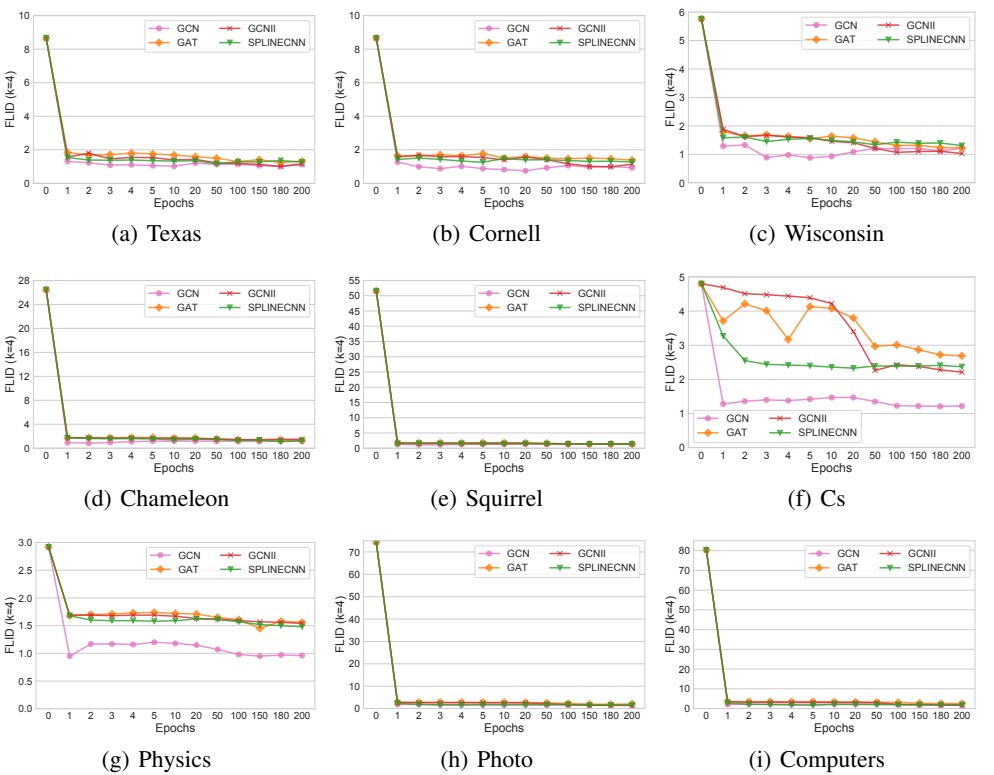

**Figure 12.** This figure shows the change of FLID (epoch 0) and RLID (epoch 1-200) across different epochs on the rest of the 9 graph datasets.

