# OpenReview forum: "Understanding Graph Learning with Local Intrinsic Dimensionality"
_ICLR.cc/2022/Conference — ICLR 2022 Submitted_

### Official Review · Reviewer_5d9E · 2021-10-31

**Correctness:** 4
**Technical Novelty And Significance:** 2
**Empirical Novelty And Significance:** 3
**Recommendation:** 5
**Confidence:** 3

**Main Review:**

Strengths
=======

The paper applies different LID measures such as FLID, SLID, and RLID in order to quantify the difficulty/complexity in learning a particular graph.  This feat is important for understanding the limitations of graph neural networks.

Weaknesses
==========

I fail to realize the innovation and practical impact of the LID measures in real-world scenarios.  In particular, I would like to have seen a proposed graph learning technique that leverages the discoveries and the importance of the analysis carried out using the LID measures discussed. Such contribution would substantially increase the relevance of this paper.  At least some preliminaries results on that direction would hugely increase the innovation of the paper.

**Summary Of The Paper:**

The paper characterizes the intrinsic dimensionality of node features, graph structures, and representations learned by GNNs via the so-called Local Intrinsic Dimensionality (LID) measure, intending that it can benefit the community in understanding the difficulty of an underlying graph learning task. In addition, estimators for Feature LID (FLID), Structure LID (SLID), and Representation LID (RLID) were introduced. This work showed that real-world graphs have much lower intrinsic dimensionality when compared to their extrinsic dimensionality.


**Summary Of The Review:**

I recommend the paper as marginally below the acceptance threshold due to the weaknesses mentioned above.

---

> ### Author Response · Authors · 2021-11-22
> **Response to Reviewer 5d9E**
>
> ###### Thank you very much for reviewing our work and the encouraging comments. Please kindly find the clarifications below to your concerns.
> ---
> **Q1.**
> I fail to realize the innovation and practical impact of the LID measures in real-world scenarios. In particular, I would like to have seen a proposed graph learning technique that leverages the discoveries and the importance of the analysis carried out using the LID measures discussed. Such contribution would substantially increase the relevance of this paper. At least some preliminaries result on that direction would hugely increase the innovation of the paper.
>
> **A1.**
> Thanks for the insightful question. Our work can help researchers understand the intrinsic complexity and learning difficulty of a graph dataset for new applications. Such knowledge could help them choose the most suitable GNN model for the new task. Meanwhile, understanding the intrinsic properties of real-world graphs (e.g., molecular interaction networks, protein interaction networks, and gene regulatory networks) itself is a challenging but important research problem. In this work, we introduced three metrics that can serve the purpose and established a set of useful understandings for graphs and graph representation learning with both synthesized and 12 real-world graphs. We believed the three LID metrics could potentially be applied to many other real-world graphs and lead to very interesting discoveries.
>
> As for using findings in this work to improve graph learning, one example is to ultilize RLID as a dimensionality regularizer to encourage the model ($f_\theta$) to learn low-dimensional (simpler) representations. The dimensionally regularized  training loss can be formulated as:
>
> $$\ell_{\rm reg}(\mathbf{x}) = \ell(f_\theta(\mathbf{x}), y) + \lambda \cdot RLID(f_\theta(\mathbf{x})),$$
>
> where $\ell$ denotes the commonly used cross-entropy loss and $\lambda$ is the coefficient of the RLID regularization term.
>
> We empirically evaluate this regularized objective on Cora, Citeseer, and Pubmed, and find that this simple exploration can indeed improve (although only slightly) existing models. As shown in the table below, it improves the performance of GCN by 0.6% on Cora, 0.4% on Citeseer, and 0.4% on Pubmed. We will explore more systematically how to use the LID measures to mitigate or solve the over-smoothing problem in our future work.
>
> | Loss | Cora | CiteSeer | PubMed |
> | ----- | ---- | --------- | -------- |
> | CE (GCN) | 84.7 | 75.4 | 88.6 |
> | CE+RLID (ours) | 85.3 | 75.8 | 89.0 |

---

### Official Review · Reviewer_XMPe · 2021-11-01

**Correctness:** 1
**Technical Novelty And Significance:** 2
**Empirical Novelty And Significance:** Not applicable
**Recommendation:** 1
**Confidence:** 5

**Main Review:**

Strengths:
1. This is an interesting topic and can help us understand many graph learning techniques.
2. The organization and presentation are good and easy to follow.


Weakness:
1. The SLID takes the distribution function as $G(y)=a^y$. However, the graphs in the real world are scale-free and directly taking the average degree for distribution definition is not reasonable.
2. Graph representation learning is a fusion of both feature information and structure information. However, I didn’t see the relationship between these three LIDs in terms of definition and experimental analysis.
3. The conclusion of Section 4.2 is wrong. The authors state that the SLID estimated on synthetic graphs goes up as the latent space dimensionality d increase in all settings. However, in some settings, the SLID first decrease and then increase as shown in Figure 4 (b). These experimental results directly disproved the conclusion.
4. The first conclusion in Section 5 is incorrect. The authors take the number of nodes and edges as the extrinsic dimensionality. As for image data, we take the number of pixels as the extrinsic dimensionality rather than the number of image samples. Similarly, this definition is correct for graph-level tasks, but for node-level tasks, extrinsic dimensions should be the dimension of features and local structures.
5. The second conclusion in Section 5 is not rigorous. The authors stated that the datasets within the same category have similar S/FLIDs and those across categories exhibit a large discrepancy. However, the FLID of Cornell is more similar to Cora rather than Winsconsin.
6. The conclusion in Section 6.1 is wrong. First, the numbers of dataset classes are not equal, and the numbers of samples belonging to different classes are not equal. It is not reasonable to compare the difficulty of classification directly according to numerical values under different task difficulties. For example, CiteSeer is with 6 classes and 75.08% accuracy while PubMed is with 3 classes and 87.6% accuracy. Apparently, CiteSeer is easier to learn, but the authors give the opposite conclusion. Second, the authors force predefined conclusions to interpret results, regardless of the real numerical relationships of the results. For example, the difference in the accuracy of Cora and PubMed is around 2%, so is the difference between Cornell and Wisconsin. But the authors state that PubMed < Cora and Cornell$\approx$Wisconsin.
7. The authors state that graphs with lower SLIDs are easier to learn in the abstract but I didn’t find the support by theory or empirical results.
8. According to the results, the SLID had no relationship with accuracy, that is to say, the structure complexity had no relationship with graph representation learning, which was inconsistent with our experience of graph representation learning.


Minors:
1. In the first line of Page 5: “a node-set E”->“an edge-set E”

**Summary Of The Paper:**

This paper study the local intrinsic dimension of graphs in terms of feature, structure, and representation.

**Summary Of The Review:**

Interesting research problem, but some definitions are unreasonable and most of the claims are incorrect.

---

> ### Author Response · Authors · 2021-11-22
> **Response to Reviewer XMPe (4/4)**
>
>
> ---
> **Q7**
> The authors state that graphs with lower SLIDs are easier to learn in the abstract but I didn’t find the support by theory or empirical results.
>
> **A7.**
> In Section 6.3 (Figure 7), we randomly drop edges from a graph, then show the decrease of SLID and increase of test accuracy within a certain range (i.e., $p \in (0, 0.5]$). We attribute the improved accuracy to the reduced intrinsic complexity and SLID of the graph (the first and second paragraph of Section 6.3). In other words, dropping edge drives down the SLID of a graph, thus making it easier to learn. We have now made it clearer in Section 6.3.
>
> ---
> **Q8**
> According to the results, the SLID had no relationship with accuracy, that is to say, the structure complexity had no relationship with graph representation learning, which was inconsistent with our experience of graph representation learning.
>
> **A8**
> In Section 6.1, we researched the relationship between classification accuracy and various graph properties. Comparing datasets within the same category, only FLID demonstrates a negative correlation with test accuracy. Even though our experiments find no correlation between test accuracy and SLID, we cannot directly draw the conclusion that structural complexity is unrelated to graph representation learning.
> The role the structure plays in graph representation learning is complex, as we mentioned above. We believe SLID should be improved in order for it to be used to uncover the underlying structural learning process, at least not averaged over the entire graph. On the other hand, if we consider the same dataset while reducing its SLID (like our DropEdge experiment in Section 6.3), we can find a clear correlation between the reduced SLID (up to a certain limit) and the improved test accuracy.
>
>
> ---

---

> > ### Comment · Reviewer_XMPe · 2021-11-29
> > **Response to the authors**
> >
> > For the added Section 7 (improving GNN performance using RLID):
> > Is there a relationship between RLID and final accuracy? The authors should give some analysis as in Section 6.1, which will make the RLID regularizer more reasonable.
> >
> >
> > For my previous questions:
> > Q1. The explanation for using the average degree as the exponent didn’t solve my concern. I think the unreasonable definition is the reason why there is no consistent conclusion about SLID in your experiment.
> > Q2. As the authors state, graph learning transforms the k-nearest neighbor information (both feature and structure) of the raw graph to find a new k-nearest neighbor graph in the representation space. It is hard to argue that this work can help understand graph learning without giving the relationship between them.
> > Q3. If the variation is caused by the estimation instability, the authors should give an analysis about why a larger radius will lead to the instability. I assume that using a larger radius means using more samples for LID estimation, which is more accurate. Besides, since most GNNs use 2-hop neighbors for information aggregation, what about r=2? Furthermore, I think the appropriate value of r is related to the node degree. The authors should give a principle for choosing r.
> > Q5. There are some value mistakes in Fig.5 (a).
> > Q6. Why the number of actual clusters (classes) will affect the properties of the neighbors found for a sample? I can’t understand the conclusion by the definitions of these three LIDs.
> > Q7. In Fig.7, the accuracy drooped when the SLID is too small. So it’s not a rigorous conclusion that graphs with lower SLIDs are easier. Moreover, using the same analysis as FLID, we can observe that there is no relationship between the SLID and accuracy.

---

> ### Author Response · Authors · 2021-11-22
> **Response to Reviewer XMPe (3/4)**
>
>
> ---
> **Q5.**
> The second conclusion in Section 5 is not rigorous. The authors stated that the datasets within the same category have similar S/FLIDs and those across categories exhibit a large discrepancy. However, the FLID of Cornell is more similar to Cora rather than Wisconsin.
>
> **A5.**
> We agree that our statement is not rigorous here. We have adjusted our claims to the following:
>
> *We find that, from the FLID perspective, the intrinsic dimensionality ranking is: Co-purchase graphs > Wikipedia graphs > Citation graphs > WebKB graphs > Co-author graphs. This indicates that the node features of one category of graphs share intrinsic properties that may be very different from that of other categories. This is not a surprise since the most discriminative node attributes of different types of graphs are indeed different. Interestingly, graphs from different categories can also have similar intrinsic dimensional properties, e.g., Cora has more similar FLID to Cornell/Texas than to CiteSeer or PubMed. From the SLID perspective, there is no consistent relationship between the two categories. It is worth mentioning that the SLID property is rather dataset-dependent, as we have shown in Figure 4 that the same type of graphs can have very different SLIDs.*
>
>
> ---
> **Q6.**
> The conclusion in Section 6.1 is wrong. First, the numbers of dataset classes are not equal, and the numbers of samples belonging to different classes are not equal. It is not reasonable to compare the difficulty of classification directly according to numerical values under different task difficulties. For example, CiteSeer is with 6 classes and 75.08% accuracy while PubMed is with 3 classes and 87.6% accuracy. Apparently, CiteSeer is easier to learn, but the authors give the opposite conclusion. Second, the authors force predefined conclusions to interpret results, regardless of the real numerical relationships of the results. For example, the difference in the accuracy of Cora and PubMed is around 2%, so is the difference between Cornell and Wisconsin. But the authors state that PubMed < Cora and Cornell ≈ Wisconsin
>
> **A6.**
> 1) Sorry for failing to understand "Apparently, CiteSeer is easier to learn". The classification accuracy of  CiteSeer with 6 classes is lower than Cora and PubMed for all 4 tested GNN models (GCN, GAT, GCNII, and SplineCNN). This clearly indicates a *higher learning difficulty*. Meanwhile, we would also like to point out that the accuracy is an averaged measure over all classes. Moreover, the LID characterization considers both the samples (explicitly) and the labels, as the labels (implicitly). This is because the number of actual clusters (classes) will affect the properties of the neighbors found for a sample. For example, in Pope et al., (2020), it shows that ImageNet (1000 classes) is harder to learn than CIFAR-10 (10 classes) as ImageNet has much higher intrinsic dimensionality.
>
> 2) It is our negligence not to use a consistent comparison standard. From Table 2,  the accuracy classification of Cornell is lower than that of Wisconsin, according to 3 out of the 4 GNN models. So, we have revised Cornell ≈ Wisconsin -> *Cornell < Wisconsin*. Note that the FLID of Cornell is lower,  our conclusion that GNNs *generally* have better performance on datasets of lower FLID still holds.

---

> ### Author Response · Authors · 2021-11-22
> **Response to Reviewer XMPe (2/4)**
>
> ---
> **Q3.**
> The conclusion of Section 4.2 is wrong. The authors state that the SLID estimated on synthetic graphs goes up as the latent space dimensionality $d$ increases in all settings. However, in some settings, the SLID first decrease and then increase as shown in Figure 4 (b). These experimental results directly disproved the conclusion.
>
> **A3.**
> We admit that our claim was not precise. We have revised our claim to: “In general, SLID is positively correlated with the latent space dimensionality of the synthetic graphs. One exception is that, in the right subfigure (SLID estimated with $r=4$), the SLID is slightly lower at $d=3$ and $p=0.3$ (the two highlighted points) than that at $d=1$. We conjecture this slight variation is caused by the estimation instability at a large radius $r=4$. A similar sensitivity has also been observed on image datasets in Pope et al., (2020). As such, we will use $r=3$ for SLID estimation in the rest of our experiments.”
>
> We would like to clarify again that the reason why we show two estimation radii $r=3$ and $r=4$ in Figure 4 (a)/(b) is that we want to test the estimation sensitivity of SLID to different $r$. We found that $r=4$ is less stable than $r=3$, as the two exception points mentioned by the reviewer. That is also why we set $k=3$ for SLID estimation for the rest of our experiments (see Table 1 and Table 3).
>
> ---
> **Q4.**
> The first conclusion in Section 5 is incorrect. The authors take the number of nodes and edges as the extrinsic dimensionality. As for image data, we take the number of pixels as the extrinsic dimensionality rather than the number of image samples. Similarly, this definition is correct for graph-level tasks, but for node-level tasks, extrinsic dimensions should be the dimension of features and local structures.
>
> **A4.**
> We agree that the number of nodes and edges may not be a good extrinsic dimensionality for node-level tasks. So, in the revised paper, we added a **3-Hop Degree** to Table 1 as the extrinsic dimensionality for node-level tasks. It takes into account the *local structure* and corresponds to the 3-hop ($r=3$) based SLID estimation.  For the node features, we believe the #features is a valid extrinsic dimensionality and so is the comparison between FLID and #features. Accordingly, we have also adjusted our analysis at the end of Section 5.
>
> ---

---

> ### Author Response · Authors · 2021-11-22
> **Response to Reviewer XMPe (1/4)**
>
> We would like to thank the reviewer for the kind review of our paper. While we accept the criticisms of our imprecise claims, we would like to argue that most of the issues are easily fixable and should not be taken to invalidate our entire work. We have revised our paper to make the analyses and conclusions more precise. Please also kindly find the following responses to your questions.
>
> ---
> **Q1.**
> The SLID takes the distribution function as $G(y)=a^y$. However, the graphs in the real world are scale-free and directly taking the average degree for distribution definition is not reasonable.
>
> **A1.**
> It is an assumption made in prior work (Ritter et al. 2018). We believe it is a reasonable assumption, as it says in (Ritter et al. 2018) *“In graphs, however, the neighborhood size often grows exponentially”*.  Please allow us to explain a little bit more about the assumption here. Given a node $v$ with degree of $a_0$, apparently $G(y=1)=a_0$. If we assume its $a_0$ neighbors connect $a_1, \ldots, a_{a_0}$ nodes respectively, $G(y=2)\approx \sum_{i}^{a_0}a_i $. We cannot specify the value of $a_i$ in the general case as the degree distributions of real-world graphs are divergent. Therefore, it is reasonable to assume $\sum_{i}^{a_0}a_i = a_0*a$, with $a$ is the average degree. Following this assumption, $G(y=2)\approx a^2$. Repeating the above steps gives us $G(y)=a^y$ by induction.
>
> ---
> **Q2.**
> Graph representation learning is a fusion of both feature information and structure information. However, I didn’t see the relationship between these three LIDs in terms of definition and experimental analysis.
>
> **A2.**
> We agree with the reviewer that graph learning is indeed a process of fusing the two types of information into the representation space, as we also mentioned in the paper. However, graph learning is a complex process that involves the transformation of the underlying k-nearest neighbor graph (k-NNG) of the raw graph to a new k-nearest neighbor graph in the representation space, i.e., neighbor nodes in the graph -> neighbor nodes in the representation space. And during this process, the raw graph structure does not change, i.e., **SLID stays the same before, during, or after learning**. So, it is hard to study the structural transformation using SLID of the original graph. We will study the SLID of the evolving underlying k-NNG in our future work. We have added this discussion to Section 6.2.
>
>
> It has been shown in a recent work *[Node Similarity Preserving Graph Convolutional Networks](https://arxiv.org/abs/2011.09643)* that there is little structural overlap between the original graph and the representations nor between the node features and the representations. Specifically, they construct k-NNGs from nodes features and the learned representations, denoted as $A_f$ and $A_h$, respectively. They then estimate the structural overlaps between the two constructed k-NNGs with the raw graph structure $A$.  We repeated the experiment and obtained the result in Table 1 below (which is consistent with the original paper). The only two substantial structural overlaps (boldfaced) occur between $A_h$ (k-NNG of the learned representation) and $A$ (the raw graph) on Cora and CiteSeer (not on two other datasets). This means that structure of the learned representation is very different from the original structure and it is difficult to analyze how feature and structure are transformed in GNNs.
>
> In Section 6.2, we have uncovered part of the relationship related to node features and the learned representations via FLID and RLID (Fig. (6)). We leave the exploration of the structural relationship via SLID to future work.
>
> Table 1. $A$: the raw graph structure (adjecency matrix); $A_f$: the k-nearest neighbour graph constructed from the node features; $A_h$: the k-nearest neighbour graph constructed from the representations (outputs of GCN's first layer). $OL(A_1,A_2)=\frac{\Vert A_1 \cap A_2 \Vert }{\Vert A_1 \Vert}$, $\Vert \cdot \Vert$: the number of non-zero (overlap) elements between the two graphs $A_1$ and $A_2$. Substantial overlaps are \textbf{boldfaced}.
>
> | Graph Pairs | Cora | CiteSeer | Actor | Cornell |
> | --------------- | ---- | --------- | ------ | ------ |
> | $OL(A_f, A_h)$ | 3.15% |  3.73% | 1.15% | 2.35% |
> | $OL(A_h, A)$ | **21.24%** | **18.77%**| 2.17% | 7.74% |
> | $OL(A_f, A)$ | 3.88% |  3.78% | 0.03% | 0.91% |
> ---

---

### Official Review · Reviewer_pEhv · 2021-11-01

**Correctness:** 4
**Technical Novelty And Significance:** 3
**Empirical Novelty And Significance:** 4
**Recommendation:** 6
**Confidence:** 4

**Main Review:**

Strength
1. This work investigates the LID of graphs, and interpreted the GNN models from the perspective of FLID, SLID, and RLID, which is interesting.
2. Authors demonstrate that FLID and SLID are good indicators of graph complexity according to node features and graph structures, respectively.
3. Experiments indicate that GNN models learn to map the raw features with high extrinsic dimensionality to low intrinsic dimensionality, and further identify the over-smoothing problem of GNNs as the collapse of the graph structure to a complete graph with $SLID=0.5$.


Weakness
The major concern is how to understand and utilize these LID metrics in graph learning. Although the investigation of FLID, SLID, and RLID is interesting, it seems that the authors do not provide any intuitive explanation of these LIDs in the paper, and these LIDs are only used to understand some operations and problems associated with GNNs. It would be great if the authors could give some more intuitive explanation of these LID metrics, and discuss how these LIDs can be leveraged in graph learning. For example, from the reviewer's perspective, one potential usage of these LIDs is to design graph adversarial attacks and more robust GNN models, as we can downgrade the performance of GNNs by perturbing graph structure or node features to change these LIDs. In addition, authors have utilized SLID to interpret the over-smoothing problem associated with GNNs, but they have not discussed how to leverage SLID to design a GNN model which overcomes the over-smoothing issue. I wonder if the authors could provide a more detailed discussion about issues related to graph learning, like the above two mentioned examples?

**Summary Of The Paper:**

In this work, the authors investigate the Local Intrinsic Dimensionality (LID), especially the feature (FLID), structure LID (SLID), and Representation LID (RLID) of a graph. Through experimental analysis, the authors demonstrate that the FLID and SLID are well correlated with the graph complexity, and real-world graphs have a much lower intrinsic dimensionality. In addition, authors also interpret the over-smoothing problem associated with GNN models from the perspective of the SLID's convergence.

**Summary Of The Review:**

In this work, the authors introduce the concept of FLID, SLID, and RLID, and investigate these metrics in graph learning. The investigation is interesting, and these LIDs are helpful in determining the graph complexity, understanding the issues associated with GNN models (e.g., over-smoothing problem). However, it seems that these LIDs are not intuitive to be explained, and this paper also does not discuss how to leverage these LIDs in graph learning. Therefore, it would be great if the authors could provide a more detailed discussion about the reviewer's two concerns mentioned above.

---

> ### Author Response · Authors · 2021-11-22
> **Response to Reviewer pEhv**
>
>
> ##### Thanks sincerely for reviewing our paper and the thoughtful comments. Please kindly find our clarifications below to your concerns.
> ---
> **Q1.**
> The major concern is how to understand and utilize these LID metrics in graph learning. Although the investigation of FLID, SLID, and RLID is interesting, it seems that the authors do not provide any intuitive explanation of these LIDs in the paper, and these LIDs are only used to understand some operations and problems associated with GNNs. It would be great if the authors could give some more intuitive explanation of these LID metrics, and discuss how these LIDs can be leveraged in graph learning.
>
> **A1.**
> Thanks for the thoughtful comment. The three LID metrics can be interpreted as measures of node discriminability to its neighbors (Houle, 2017). They can be utilized as regularizers or supervision signals to guide graph learning towards more locally discriminable (low LIDs) representations. For example, we could use an RLID regularizer to encourage the model ($f_\theta$) to learn low-dimensional representations, as follows:
>
> $$\ell_{\rm reg}(\mathbf{x}) = \ell(f_\theta(\mathbf{x}), y) + \lambda \cdot RLID(f_\theta(\mathbf{x})),$$
>
> where $\ell$ denotes the commonly used cross-entropy loss and $\lambda$ is the coefficient of the RLID regularization term.
>
> We empirically evaluate this regularized objective on Cora, Citeseer, and Pubmed, and show that it can improve existing models. As shown in the table below, it improves the performance of GCN by 0.6% on Cora, 0.4% on Citeseer, and 0.4% on Pubmed. We will add more results later on. Note that this is just a simple attempt so the improvement is not so significant. We believe that a more advanced strategy is to exploit the FLID or SLID of each node as \emph{targets} of the dimensionality regularizer to prevent the collapse of the intrinsic structure in the presentation space and thus the over-smoothing problem. We will leave these explorations to our future work.
>
> Meanwhile, the three metrics can also be exploited to detect graph adversarial examples or regularize GNNs to be robust to adversarial examples like the LID detector developed in the image domain by (Ma et al., 2018a).
>
>
> | Loss$\downarrow$, Dataset$\rightarrow$ | Cora | CiteSeer | PubMed |
> | ----- | ---- | --------- | -------- |
> | CE (GCN) | 84.7 | 75.4 | 88.6 |
> | $\ell_{\rm RLID}$ ($\lambda=1$) | 85.3 | 75.8 | 89.0 |
>
> ---

---

### Decision · Program_Chairs · 2022-01-20

**Decision:**

Reject

**Comment:**

The paper investigates the interesting problem of the local intrinsic dimension (LID) of graphs, and interpreted the GNN learning from Feature LID (FLID), Structure LID (SLID), and Representation LID (RLID). The concepts are novel but the paper needs better insights on how LID can improve graph learning and stronger empirical evidence to support their claims.